# Spatial Determinants of Real Estate Appraisals in The Netherlands: A Machine Learning Approach

**Evert Guliker** [1,2,3]**, Erwin Folmer** [1,4,*] 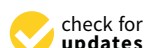 **and Marten van Sinderen** [2]

1 Faculty of Behavioural, Management and Social Sciences (BMS), University of Twente, 7522 NB Enschede, The Netherlands; b.e.guliker@alumnus.utwente.nl
2 Faculty of Electrical Engineering, Mathematics and Computer Science (EEMCS), University of Twente, 7522 NB Enschede, The Netherlands; m.j.vansinderen@utwente.nl
3 Stater N.V., 3826 PA Amersfoort, The Netherlands
4 Kadaster, 7311 KZ Apeldoorn, The Netherlands
* Correspondence: erwin.folmer@utwente.nl

**Abstract:** With the rapidly increasing house prices in the Netherlands, there is a growing need for more localised value predictions for mortgage collaterals within the financial sector. Many existing studies focus on modelling house prices for an individual city; however, these models are often not interesting for mortgage lenders with assets spread out all over the country. That is why, with the current abundance of national geospatial datasets, this paper implements and compares three hedonic pricing models (linear regression, geographically weighted regression, and extreme gradient boosting—XGBoost) to model real estate appraisals values for five large municipalities in different parts of the Netherlands. The appraisal values used to train the model are provided by Stater N.V., which is the largest mortgage service provider in the Netherlands. Out of the three implemented models, the XGBoost model has the highest accuracy. XGBoost can explain 83% of the variance with an RMSE of €65,312, an MAE of €43,625, and an MAPE of 6.35% across the five municipalities. The two most important variables in the model are the total living area and taxation value, which were taken from publicly available datasets. Furthermore, a comparison is made between indexation and XGBoost, which shows that the XGBoost model is able to more accurately predict the appraisal values of different types of houses. The remaining unexplained variance is most probably caused by the lack of good indicators for the condition of the house. Overall, this paper highlights the benefits of open geospatial datasets to build a national real estate appraisal model.

**Keywords:** real estate values modelling; housing market; housing price; real estate appraisals; hedonic model; extreme gradient boosting; geographically weighted regression; The Netherlands

## 1. Introduction

In the Netherlands, it is mandatory to get an appraisal by a certified appraiser when taking out a mortgage, as mandated by the Authority Financial Markets (AFM) [1].These appraisals play an important role in applying for a mortgage. In mortgage lending, the ratio between the borrowed sum and the collateral value is called Loan-to-Value. Loan-to-Value and Loan-to-Income are the two most important determinants of how much money can be borrowed. They serve as a good indicator for the risk of the mortgage lender [2] and protect people from taking on a mortgage they cannot afford.

Appraisals can be wrong. For example, in 2018, DNB, the central bank of the Netherlands, released a critical report about the quality and independence of Dutch housing appraisals [3]. They concluded there is a structural over-appreciation by appraisers, based on 95% of all appraisals being equal to or higher than the sale price (in the observed period). Striving for accurate appraisals is important not only for management of the mentioned risk but also to build trust between the home buyer and the financial sector, which is beneficial to society.

We can distinguish between traditional appraisals and model-based appraisals. With traditional appraisals, an appraiser visits a house to evaluate its condition. The intrinsic characteristics of the house determine a large part of its price; examples include the number of bedrooms, amount of living space, presence of a garden or garage, and presence of solar panels. The appraisers weight these factors and compare the sale prices of houses with similar characteristics. Ultimately, the appraiser tries to make an objective estimation of the property value. Traditional appraisals are accurate but time consuming and therefore expensive.

In contrast, model-based appraisals make an automated model-based estimation of the price of a particular house by using data on similar houses that have been previously sold. One of the advantages of model-based appraisals over traditional appraisals is that they are cheaper. However, the accuracy of model-based appraisals depends on the amount of data on similar houses that can be used as a reference.

Hedonic pricing models, which estimate house prices using quantitative data about the house characteristics, location, and the supply versus demand, can be used to improve model-based appraisals. The literature has shown that for many cities, e.g., London [4], Rotterdam [5], Leipzig [6], and Singapore [7], the house prices can be estimated using these types of models. However, many of these models focus on a single city within a country.

Model-based estimations based on hedonic pricing models are already being used in practice as an alternative to the traditional appraiser. In the Netherlands, a notorious example is the WOZ-waarde, which is a taxation value established by the government. At its core, the WOZ-waarde comes from matching the sale prices of houses with similar characteristics [8]. Similarly to a hedonic model, it uses the characteristics and location of the house to make a prediction. These data come from official registries from the Kadaster, which is the independent administrative body in the Netherlands for maintaining property registries, such as the base registry of addresses and buildings (BAG) [9]. In actuality, the model is more complex than a hedonic price model. It uses many extra layers for improving and validating the accuracy of the model. For example, they conduct samples of physical appraisals for very unique houses to ensure validity. In addition, satellite pictures are used to check whether houses have the registered physical properties (e.g., a house owner may have built a house extension or swimming pool, which increases the property's value). A homeowner is able to get a report about the WOZ-waarde of his home. This report contains houses similar to that of the homeowner, which are used to derive the WOZ-waarde.

The WOZ-waarde serves as an indication of value for the property, which is used by the municipality for taxation. It is simply impossible for municipalities to appraise every single house through house inspections on an annual basis. Many insurance companies and mortgage lenders are in the same boat: the costs to conduct a traditional appraisal for each and every house in their portfolio is simply too high. However, there is a limitation on the use of the WOZ-waarde, as these data cannot be requested in bulk for each individual house, without sufficient legal grounds. Therefore, many mortgage lenders and insurance companies opt to adjust the house values in their portfolio with national indices to re-evaluate the house prices. The drawback of indexation is that it generalises different factors that determine house value into a single index. Consequently, houses can still be over- or undervalued, for example if price growth rates differ for different regions, location characteristics, or house types.

A commercial example of a (hedonic) house price model is Calcasa [10]. Calcasa, a fintech company, puts itself in the market with their own real estate valuation model, which is certified by rating bureaus such as Moody's, Fitch Ratings, and Standard & Poor's. They target insurance companies and mortgage providers to provide model-based appraisals for their portfolios. Unfortunately, as this is their business model, it is unclear what exact model they run. However, Calcasa uses housing characteristics combined with historic sales data for their model, which is similar to what the WOZ-waarde model uses.

All in all, from these examples, it can be seen that there definitely exists a market for house price models in the Netherlands. All these models seem to rely on systems that try

to match the sale prices of similar houses based on their characteristics. These sales data are the key starting point for all models. If enough sales data are present, the most difficult challenge is collecting as much accurate data about a house as possible. The main physical characteristics, as well as neighbourhood characteristics, are publicly available through the Dutch Kadaster and Central Agency for Statistics (CBS) respectively. In the end, whoever has the most, but also accurate, data will ultimately be able to make the best prediction.

To the best of our knowledge, no hedonic pricing models currently exist that can conduct house price estimations across different cities. The goal of this paper is to investigate the model-based appraisal of real estate using hedonic pricing across cities and publicly available data. We want to compare different machine learning (ML) approaches to produce hedonic pricing models and evaluate these on basis of accuracy, cost, speed, and data requirements.

The research questions that we address to achieve this goal are as follows:

1.  Which ML approaches are currently used for hedonic pricing, and how do they perform?
2.  Which factors are significant for price differences between houses across cities?
3.  Which data are available about these factors?
4.  How can we construct a method for hedonic pricing across different cities using the obtained insights?
5.  What are the results of applying this method with a realistic dataset?

The remainder of this paper is organised as follows: Section 2 gives a background overview of traditional price indexation and four hedonic pricing models for real estate appraisals: (1) linear regression, (2) geographically weighted regression (GWR), (3) multi-scale GWR (MGWR), (4) extreme gradient boosting (XGBoost), as well as the variables commonly used in these models. Section 3 introduces the data sources and model metrics used in this paper to build the models. Section 4 evaluates the models and their drawbacks, as well as compares their performance against traditional indexation. Section 5 discusses the implications of the model results. Finally, Section 6 presents the conclusions to the research questions and areas for further research.

## 2. Background

This section discusses the benefits and limitations of two approaches for estimating house prices: price indices and hedonic pricing models. Simultaneously, the price index and other house price indices of the Netherlands are explored to show developments in the Dutch housing market. Furthermore, this section evaluates both two practical models as well as four state-of-the-art models commonly used in the literature for hedonic price models: linear regression (LR), geographically weighted regression (GWR), multi-scale GWR (MGWR), and extreme gradient boost (XGBoost). Finally, an overview is given of common features for such hedonic price models. This overview is divided into three categories: market characteristics, location characteristics, and intrinsic characteristics of the house.

### 2.1. Dutch House Price Indices And the Repeat-Sales Model

Price indexation is a method for calculating a normalised average price increase for different types of goods. Four common methods to calculate an index are as follows: (1) Paasche index, (2) Laspeyres index, (3) Lowe index, and (4) Fisher index. Every index aims to give a good indication for the price change during a specific interval of time. A price index is often used to estimate the present value using a historic known value. This process is called indexation. In the case of house prices, the current value of a house can be estimated by using a sale price from the past and indexing it using a house price index.

For the Netherlands, a notable house price index is calculated by the Kadaster. The Kadaster is the Dutch land registry and mapping agency. It maintains the official registry of properties and land ownership in the Netherlands. This registry is called the Base-registry Addresses and Buildings (BAG). The house price index, together with other statistics related

to the Dutch housing market, are presented in a publicly available dashboard, which is updated every month.

The Kadaster index is calculated using a weighted repeat-sales model [11]. The four aforementioned methods for calculating price indices require multiple sales of the same good, in the desired time span, for an accurate index. This means multiple sales of the same good per year for a yearly based index. However, this is not the case for houses, which often do not get traded for decades. The repeat-sales model is developed to specifically circumvent this issue.

The repeat-sales model averages the change in sale price for a single good between two different moments in time [12]. In case of house prices, it averages the change in price for the same house that has been sold in separate years. Inevitably, a prerequisite for this model is the need for at least two separate sales dates for every unique house. The repeat-sales model is not only used to calculate house prices but other infrequently traded goods such as collectables (e.g., pieces of art). The weighted repeat-sales model expands on the model by having more frequently traded houses contribute less to the total average than houses traded over a larger span of time. This avoids bias towards more frequently traded houses.

Additionally, the Kadaster house price index consists of two unique refinement levels: one is for the different provinces of the Netherlands (Table A1), the other is for six different types of housing (Table A2). Both indices are based on all real estate transactions of the last twenty years (2000–2020), with 2015 as the base year. While the house prices follow the same trend, the small differences over many years add up to a significant differences over time [11]. The largest increase is seen in Noord-Holland, where prices have risen up to 76.70%, which is twice as high as that of 38.16% in Limburg (as seen in Table A1). For different types of houses, the difference is also statistically significant, as proven in [11]. Considering these facts, it can be concluded that additional factors are needed in order to model the house prices on a more localised scale for the Dutch housing market.

In the end, indexation provides a reasonable estimation for house prices but only on a global scale. In a local model, when one wants to estimate the current value of a specific house, an index is likely to give a 'good enough' estimation. For a single house, an index cannot quantify the exact price change, since it is based on the average price change of a larger sample. Including different factors to compose more indices improves the accuracy. Despite this, the biggest downside still remains. Indices rely on large samples of the total transactions to be reliable. By using regression, hedonic price models are a valid alternative when a large data sample is unavailable.

### 2.2. Hedonic Price Models

Hedonic pricing states that the price for a product is an aggregation of prices a buyer is willing to spend for individual characteristics of the product. For a house, these characteristics range from intrinsic characteristics (e.g., number of rooms) to location characteristic (e.g., access to amenities) as well as market characteristics (e.g., supply of houses in the area) [13]. Correspondingly, house prices reflect macro-economical changes in the wishes and values of society. As such, house prices play a versatile role in quantifying the price of intangible goods such as clean air [4], the presence of green space [14], and accessible infrastructure. Hedonic price models use different types of regression models to estimate the price and weight of each characteristic. The four types of regression models used in recent research for hedonic house price estimations are: (multi) linear regression, geographically weighted regression (GWR), multi-scale GWR (MGWR)—an improvement upon GWR—and extreme gradient boost (XGBoost).

### 2.3. Linear Regression (LR)

Linear regression (LR) models the change in a dependent variable based on a linear relationship to one or multiple independent variables. Using ordinary least squares, the influence of each feature is described by a single coefficient. Research successfully shows that

linear relationships exist between house prices and the living surface area of a house [15]. Furthermore, many other intrinsic characteristics such as the number of bedrooms [16] and the amount of garden space [14] show an underlying linear contribution to the price of a house. The advantage of the linear regression model lies in its simplicity to have the same response for all data points. As a result, linear regression models are generally less prone to over-fitting the dataset.

Conversely, the simplicity of linear regression models is also their downfall when it comes to modelling more complex phenomena such as house prices. In practice, many other factors that play a role in house prices also show non-linear relationships [5]. For example, an additional room has a larger influence on the value of an apartment than it has for a detached home. This can be resolved by breaking down the non-linear relationship into a linear relationship by including another feature, in this case the type of house. However, it is often the case that the non-linear relationships simply cannot be broken down into linear relationships through the inclusion of additional features.

Finally, linear regression models are argued to not be a good estimator for house prices due to the lack of modelling a spatial component [16]. House prices for the same type of house in Amsterdam vary wildly from those in Groningen [17]. Both on a national level, as well as city level, the price of the same house is often different. This is because of spatial heterogeneity, meaning the value of a variable varies across space. Not considering spatial heterogeneity in the model causes spatial non-stationarity. Spatial non-stationarity is the name [18] for the situation in which a global model, such as linear regression, is unable to accurately predict the outcome due to location playing a role.

One way to mitigate the spatial non-stationarity problem is to group observations through the use of a dummy variable, such as the inclusion of zip codes [19] or distance to the centre of the city [20]. Furthermore, it is argued that through quantifying enough features, it is possible to distinguish regions [21]. Nevertheless, the drawback of quantifying more features is that it is very data intensive to make reliable distinctions. Despite all this, the model still ignores the spatial dependence of houses located nearby, which has been proven to be statistically relevant when modelling house prices. All in all, the lack of spatial component and subsequent decrease in model accuracy cannot be significant when looking only at the individual characteristics of houses in a neighbourhood or city.

### 2.4. Geographically Weighted Regression (GWR)

Geographically weighted regression (GWR) is a parametric model based on traditional linear regression but also takes into account the spatial heterogeneity to avoid the problem of spatial non-stationarity. Similar to linear regression, GWR gives each independent variable an estimated coefficient; however, the coefficient varies spatially depending on near data points [18]. Which points are considered near enough and the weight each point gets assigned is defined through a kernel function. GWR has proven beneficial for better accuracy based on both intrinsic characteristics [5] and location characteristics [6].

For spatial analysis such as GWR, it is important to know about spatial autocorrelation. Spatial autocorrelation is most famously described in a quote by Tobler, which is also known as the First Law of Geography: "Everything is related to everything else, but near things are more related than distant things" [22]. More formally, spatial autocorrelation is the correlation between data points of nearby locations in space. Commonly used statistics for determining spatial autocorrelations are Moran's I and Geary's C test statistics. Spatial autocorrelation can be an indication of missing a dependent variable. In turn, this means that the model is wrongly specified, leading to results that can be statistically invalid.

The kernel function plays an important role in how the model weights each of the coefficients. Two main types of kernel functions exist: (1) fixed, which considers data points in a fixed radius, and (2) adaptive, which considers a fixed amount of neighbours. An adaptive function automatically adjusts its bandwidth to always include the same number of data points. This makes it ideal for spatial datasets, which are not uniformly distributed spatially. The most commonly used kernel function across the identified literature in real

estate pricing is the adaptive Gaussian kernel, which considers all observations but the weight tends towards zero the farther away an observation is [5–7,23]. The kernel function of the GWR model can be optimised through usage of the golden search method and cross-validation. The step of kernel function optimisation is crucial, as a randomly chosen kernel function decreases the accuracy of the model.

A downside of the GWR model is the fact that the kernel function is forced to have the same bandwidth for all variables. The bandwidth is the amount of data points that are weighted in the kernel function. Different variables might exert influences over larger or smaller areas. In this case, it is wrong to assume a constant bandwidth. Some effects can only be related to influences of other houses in the same neighbourhood, while others are globally influenced by all data points in the city. This simplification of reality sparked the creation of a new variation upon GWR that does include variable bandwidths, which is called multi-scale geographically weighted regression.

*2.5. Multi-Scale Geographically Weighted Regression (MGWR)*

Multi-scale geographically weighted regression (MGWR) introduces variable bandwidths for each of the coefficients [24]. Despite the first publication in 2017, this model has seen fewer studies than GWR, both overall as well as in the context of house price estimations. This can be due to the fact that popular spatial analysis tools, such as ArcGis, do not yet have a built-in MGWR analysis, only for GWR. The recent release together with no major support of spatial analysis tools has meant that less research has been conducted on MGWR as compared to GWR.

Nevertheless, research has shown that MGWR often offers an improvement over GWR [24]. However, the described improvements vary across studies. These differences are sometimes too small to be statistically significant. As seen in [25], the explained variance ($R^2$) shows a minor increase of 0.05 (10% improvement) when switching from GWR to MGWR. Furthermore, a recent study into prices of AirBnB rental prices also had a 0.10 improvement with the use of MGWR versus GWR [26]. Overall, research [25,26] agrees that the different local and global influences of variables are the main benefit of MGWR over GWR.

*2.6. Regression Trees and Extreme Gradient Boost (XGBoost)*

Although with (M)GWR, the coefficients can vary spatially to model positive influences in one location as well as negative influences in another location, they still rely on linear relationships to perform regression analysis. An alternative to this is a decision tree model, which is able to model non-linear behaviour. Commonly used for classification, decision trees can also be used for regression, often called regression trees in that scenario. Gradient boosting is a technique that uses the ensemble learning of many weak prediction models to make better prediction than using a single tree. Finally, extreme gradient boost (XGBoost) is a library that implements this gradient boosting for tree models in a way that is fast and efficient.

XGBoost also has applications for predicting house prices. It has been used to model the Boston housing dataset with a mean absolute percentage error of less than 5% [27]. This dataset is a popular dataset for Kaggle competitions to compare the performance of various machine learning models. Similar to the Boston dataset, most other applications of XGBoost also focus on modelling house prices based on intrinsic characteristics of the house itself [28]. Overall, this makes XGBoost another prime candidate for a hedonic pricing model that can also capture non-linear relationships.

*2.7. Features for House Price Estimations*

Based on the analysed studies and practical applications for hedonic pricing models, a list of characteristics is identified and divided into three categories: market characteristics, location characteristics, and intrinsic characteristics of the house. The two most important categories are the intrinsic and location characteristics of the house, since the market charac-

teristics are global influences impacting all houses. Nevertheless, the market characteristics have been included for the sake of completeness. This overview is based on the overview of hedonic model variables of Zhou et al. [16]. However, this overview focuses mainly on variables that have also been included in geographically weighted regression models.

The market characteristics are identified as global influences on the entire housing market. One large market influence is national policies, such as the recent abolition (January 2021) of the transfer tax for starters in the Dutch housing market. These national policies often have an uniform impact on all housing prices [21]. Another global influence is the mortgage interest rate. A lower interest rate leaves the home buyer with more money to spend. As a result, this often drives up house prices. Since market characteristics are global influences, it does not explain the spatial variance in house prices. As such, these variables do not belong in a geographically weighted regression model. Nevertheless, they play a crucial role in explaining the temporal difference in houses prices, as they do play a role when looking at the growth of house prices on a yearly basis.

In contrast, intrinsic characteristics are the biggest differentiating factors for house prices [4,29,30]. As such, they are also by far the most used variables for hedonic pricing models [16]. Not only in the literature, but also in practical applications, such as the Dutch taxation model, these variables play the dominant role. The largest influences are the living area and volume [16], which are commonly followed by the amount of garden space. Amenities such as a garage and multiple bathrooms also contribute to higher house prices. The build year can serve as a moderate indicator of energy efficiency and state of maintenance; however, it does not always depict the true condition of the house. Old houses are likely renovated once in their lifespan, so other features such as an energy label are needed. Furthermore, older buildings can also be cultural heritage, which can result in higher prices for older buildings due to their significant historic value as stated in [5]. The complete overview of all variables is given in Table 1.

The largest downside of intrinsic characteristics is that open data about these characteristics are hard to come by. Most data of real estate agencies are either protected or can only be bought. Despite this, good public national sources for house characteristics do exist in the Netherlands. The Kadaster provides basic information about every house including year of construction and living area.

In the literature, the majority of the GWR models for house pricing focus on modelling only intrinsic characteristics based on data gathered from real estate marketplaces or real estate agencies [5,31–33]. However, research [4,7] also shows that characteristics about the location/neighbourhood of the house also contribute to the house price. According to [4], the location/neighbourhood accounts for 15% to 50% of the total house price. As such, even when little data are available about each specific house, a more local estimation can still be performed using location characteristics.

**Table 1.** Identified location characteristics influencing house prices.

| Characteristic | Influence | Sources |
| --- | --- | --- |
| Year of construction | Positive/Negative | [5,16,34] |
| Living area | Strongly positive | [5,13,16] |
| Type of housing | Positive | [5,13,16] |
| Garden space/presence of garden | Positive | [13,16] |
| # of rooms (bedrooms, bathrooms) | Positive | [13,16] |
| Presence of facilities (shower, lift, garage, etc.) | Slightly positive | [13,16] |
| Furnished | Slightly positive | [13,16] |
| Energy Efficiency | Slightly positive | [5] |
| Sustainability measures | Slightly positive | [5] |

In this paper, location characteristics refer to features derived from the type of neighbourhood and the presence of nearby buildings. For example, nearby access to convenience stores, recreation, and parks all have positive influences on house prices [19]. This agrees

with bid rent theory, which states that rent for housing gets higher the closer the house is to the central business district.

Similarly, accessibility plays another role in the price of a house. Travel time to certain locations such as the central business district can be a better indicator than the distance. However, not all forms of transport are a positive influence. The proximity of a highway has a larger detrimental effect. The effect of the noise disturbance is greater than the impact on the better accessibility of other cities. Views also play a role; an outlook on a river, lake, or sea can have positive influences, whereas windmills and high-rise buildings have detrimental effects.

Lastly, there are socio-economic indicators for a neighbourhood that also relate to house prices. A higher average household income is most often found in areas with more expensive housing. Crime rate often has a negative impact on house prices. When researching these relationships, it is important to discover if there actually is a casual correlation or not. Overall, the location characteristics have a less pronounced effect than the most intrinsic characteristics, as the value associated with each of them varies on a personal basis, yet they can still provide large insights into why certain houses have higher house prices than others. A summary of the location variables is given in Table 2.

**Table 2.** Identified location characteristics influencing house prices.

| Characteristic | Influence | Sources |
|---|---|---|
| Household income | Strongly positive | [7,18] |
| House shortage | Strongly positive | [35] |
| Notable view (sea, lake, park) | Strongly positive | [33] |
| Time to travel or distance to city centre | Strongly positive | [14,19] |
| Proximity to place of worship | Positive/Negative | [5,36] |
| Distance to highway | Negative | [37] |
| Distance to heavy industry | Negative | [37] |
| Presence to high rise/view obstruction | Negative | [16] |
| Crime rate | Negative | [19] |
| Unemployment rate | Slightly negative | [18] |
| Population density | Positive | [35] |
| Presence of cultural landmarks | Slightly positive | [18] |
| Birth surplus | None | [36] |

## 3. Data and Methods

In this study, we build three hedonic pricing models to predict appraisal values for houses in the Netherlands based upon the models and variables discussed in the previous section. The chosen models are (1) LR, (2) GWR, and (3) XGBoost. Each model is applied to real-world appraisal data provided by Stater N.V., which is the largest mortgage service provider in the Netherlands. The models use data from 2018 and 2020 for five large selected municipalities spread out across the Netherlands, namely Rotterdam, Amsterdam, Eindhoven, Amersfoort, and Groningen. The assumption is made that this dataset provides sufficient variety to train the model for any particular city in the Netherlands. Finally, this section concludes with an overview of the explanatory variables and model parameters that are optimised.

### 3.1. Model Metrics

The end goal is to discover if the house and location characteristics allow for reasonable predictions of appraisals, and if this is a better approach than traditional indexation. The three models are evaluated using quantitative as well as qualitative metrics.

### 3.1.1. Quantitative Metrics

The quantitative metrics are based upon common accuracy performance metrics for machine learning models. First, the $R^2$ serves as a measure for goodness of fit. Secondly, the

prediction error is quantified by the root mean squared error, or RMSE. The RMSE weighs large errors more heavily than smaller ones by squaring them. This is the metric that is often used to optimise regression models. Additionally, the MAE is calculated, which is the absolute mean average error. The MAE is always lower or equal to the RMSE, as it does not put a heavier weight on larger absolute errors. Finally, the mean absolute percentage error, or MAPE, gives the relative error. This is helpful, as house prices range from €150,000 to over a million, and as such, more expensive houses with absolute larger errors do not skew the accuracy of the model.

### 3.1.2. Qualitative Metrics

A slightly more accurate model is not necessarily better if the maintainability of the model has much higher costs. The qualitative metrics aim to provide better insight into the operational costs to implement the model and keep the model up-to-date. The two main metrics here are (1) model implementation time: how much time/effort it would take to replace the current model, (2) model upkeep: how much time needs to be spent on keeping the model up-to-date and running (loading new data and training the model).

### 3.2. Exploration of the Response Variable

Each mortgage application in the Netherlands needs an official appraisal by a certified appraiser. The appraisal value, expressed in euros, is what is used as an indication of the property value. This is used as the response variables for the models. The total number of real estate appraisals per year is given in Figure 1a. It highlights that the total amount of appraisals varies per year. For example, around the financial crisis of 2007–2008, there were a lot less mortgage applications. On the other hand, recent years have more mortgage applications due to the increasing demand on the Dutch Housing market.

Additionally, Figure 1b shows that the number of appraisals varies per municipality. This appears to be roughly correlated with the population density of the Netherlands, where larger municipalities have more appraisals. Figure A1 in the appendix shows that this distribution remains similar across years. In years with few mortgage applications, such as 2008, many smaller municipalities only have around 300 appraisals, which is only a small fraction of their total amount of houses. For these regions, it is harder to make accurate predictions. Instead, we focus on five large municipalities, namely Rotterdam, Amsterdam, Eindhoven, Amersfoort, and Groningen. If the models make predictions with good accuracy for these five regions, then they already cover a large percentage of Stater's dataset.

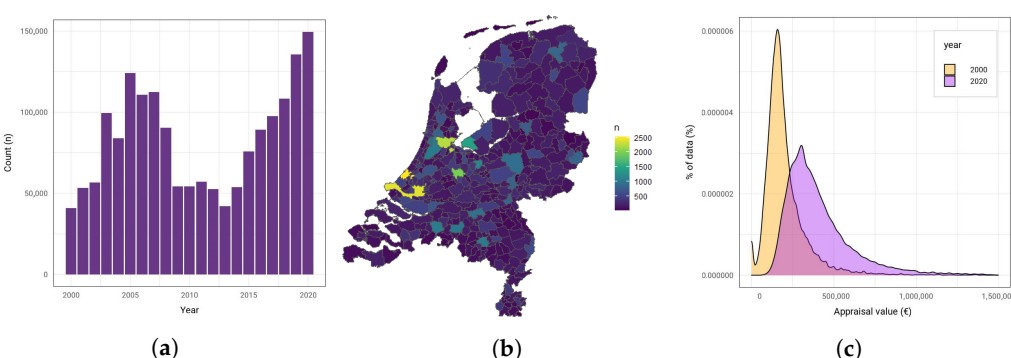

(a)　　　　　　　　　　(b)　　　　　　　　　　(c)

**Figure 1.** Exploration of the residential real estate appraisal dataset of Stater N.V. (**a**) Appraisals per year (2000–2020). (**b**) Records per municipality (2020). (**c**) Increase in average appraisal value, (Amersfoort, 2000 & 2020).

Average houses prices vary across the Netherlands. Similarly, the average appraisal value of the dataset also varies per municipality and also in time. For the appraisals values of 2000 and 2020, an increase in the number and average appraisal value can be seen between 2000 and 2020 (Figure 1c). This means that a complete prediction model for appraisal values would need to discern the differences both in time and regional location.

However, the goal of this paper is not to explain the differences between years and predict future appraisal prices for houses, which is a more difficult task requiring a different approach. For mortgage services, the current value of a mortgage collateral matters the most. As such, it is not a problem to only train the models for a specific year. In this paper, the models are trained on data from 2018 and 2020. 2020 is chosen because this is the most recent complete year. Additionally, 2018 is chosen to validate the model for a different year with less appraisals. For 2018, the number of appraisals for these 5 municipalities is summarised in Table 3.

**Table 3.** Descriptive statistics for appraisal values per municipality (2018).

| Municipalities | Samples | Mean | Std Dev. | Min | Max |
|---|---|---|---|---|---|
| Amersfoort | 1494 | €319,400 | €62,744 | €58,800 | €1,250,000 |
| Amsterdam | 5084 | €451,650 | €84,992 | €81,000 | €1,500,000 |
| Eindhoven | 1845 | €278,800 | €58,421 | €75,000 | €1,155,000 |
| Groningen | 1160 | €222,610 | €49,143 | €45,000 | €955,000 |
| Rotterdam | 3011 | €254,930 | €53,329 | €55,000 | €875,000 |

*3.3. Exploration of the Explanatory Variables*

The appraisal dataset contains additional data about the house type (apartment or family home) and the presence of a garage or parking space. These categorical variables are transformed using one-hot encoding, since the models can only accept numerical data. Furthermore, four datasets are used to collect more information about the houses and their location. They come from three parties: the Dutch cadastral registry (Kadaster), the Dutch Central bureau of Statistics (CBS), and the Netherlands Enterprise Agency (RVO); see Table 4.

**Table 4.** External data sources for additional housing characteristics.

| Dataset Name | Contents | Joined Using | Source |
|---|---|---|---|
| BAG: 'Addresses and Buildings key register' | Geo-coordinates, build year, surface area | Address | Kadaster [9] |
| DKK: 'Digital cadastral map' | Land lot area | BAG-VBO-ID | Kadaster [38] |
| CBS Square statistics | Variables for areas of $100 \times 100$ m and $500 \times 500$ m | Geo-coordinates | CBS [39] |
| EP-Online | Energy labels | BAG-VBO-ID | RVO [40] |

As mentioned in Section 2, the Kadaster maintains the central registry related to land ownership in the Netherlands. Their base registry of addresses and buildings (BAG) [9] provides geo-coordinates for each valid address in the Netherlands as well as total living area and the house's build year. The BAG data is joined via the address—a combination of ZIP code, street name and house number—from the appraisal dataset.

In addition to information about the actual houses, the Kadaster also has information about the boundaries of all land lots in the Netherlands, which are stored in the DKK [38]. As the literature has shown, lot area is less important than the living area but still influences house prices. Especially in the city centres, more garden space is valuable. For this research, the Kadaster has provided the 'Location Cadastral Object' (LKO) table, which links land lots from the DKK to the buildings from the BAG. The land lot data are joined using a building ID that is available in the BAG.

All in all, after joining and computing the combined surface area of all land lots, on average, 69.3% of all family homes have an associated land lot area. For all apartments that are missing a land lot, a zero is filled in, as apartments generally do not have a land lot. A scatter plot of the Kadaster variables is given in Figure A2a, which shows a strong relationship between the appraisal value for both the living area and the land lot area. Finally, the overall percentage of missing records for this variable is summarised in Table 5 under 'Land lot area'.

The next dataset is the so-called 'Square statistics' from the CBS [39]. The CBS publishes many sociographic and demographic variables about the entire Netherlands. They publish

these data for different levels of resolution. From the highest resolution to the lowest resolution, the following sets are published: full postal code (PC6), 100 × 100 m tiles, 500 × 500 m tiles, 4-character postal code (PC4), and neighbourhoods and city blocks. The neighbourhoods and even municipalities can merge, split, or change borders. In this paper, the 100 × 100 m and 500 × 500 m datasets are used. One of the main advantages of the tile dataset is that their size and geographical position remains constant throughout the years. Figure 2 gives an example of three variables for Amersfoort (2018).

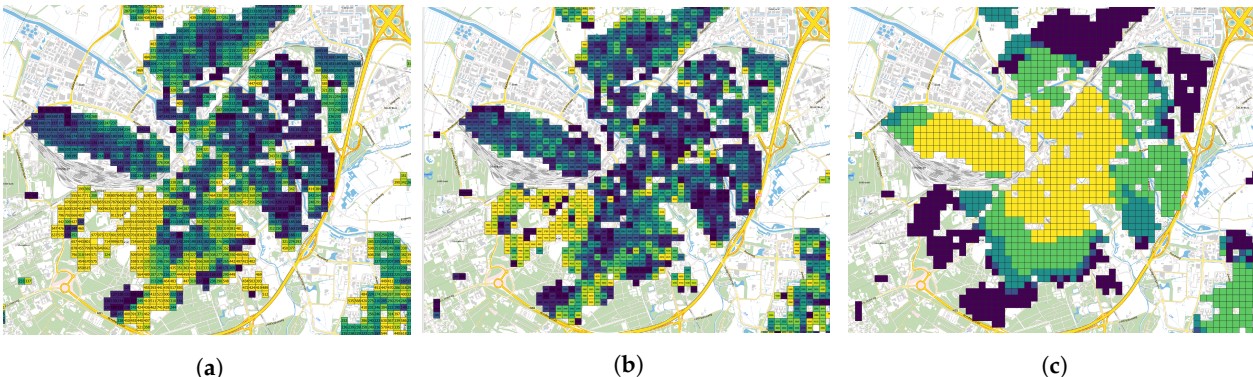

| (a) | (b) | (c) |

**Figure 2.** Various CBS 100 × 100 m statistics (Amersfoort, 2018). (**a**) Taxation value (WOZ-waarde) (€1k). (**b**) Electricity usage (kWh). (**c**) Nearest cafe (km).

Joining the tile dataset is possible using the geo-coordinates from the BAG. However, not every house lies within a tile. The main reason is that tiles with less than 5 households have their values censored due to privacy reasons. This problem was mainly an issue with demographic variables, such as the number of people aged 0–14 years, 15–24 etc. and the average taxation value (WOZ-waarde). It is not possible to mix and replace the 100 m tiles with the 500 m tiles for absolute values, such as the number of people aged 0–14 years. On the other hand, if the value is an average, it is possible to use the 500 m tiles, since the 500 m tiles will just give a more generalised average of a larger sample. For average income and the average taxation value, Table 6 quantifies how large the subset of data is that has the missing values of 100 m tiles replaced with 500 m tiles; this is on average 5% of the total number of observations.

Furthermore, inside the CBS dataset, there are many variables that list the distance to nearest 'X' or the amount of 'Y' within a certain radius of the tile. These are abbreviated respectively with 'AFS' and 'AV##' (where ## specifies the radius in km). The X and Y refer to facilities such as grocery stores, cafes, swimming pools, hospitals, cinemas, and more. The 'distance to' and 'amount within radius' variables that describe the same type of building end up being highly correlated. As such, only the 'distance to...' variables are included. To summarise, the total variable overview of Table A4 in the appendix lists the descriptions of all variables and which tile set they use (variable names ending in _100 or _500).

Additionally, based on the geo-coordinates from the BAG, it is possible to calculate the distance to the city centre for each house. The coordinates of the city centres are manually determined using Google maps. For the five municipalities in this research, this is still doable by hand. However, for the entire Netherlands, a different solution must be found. The resulting variable is called 'dist_centre'. In the end, the distance to the city centre variable turns out to also correlate with the CBS distance variables. For example, as can be seen in Figure 2c, there is a relationship between the distance to cafe and the distance to the city centre of Amersfoort. For linear regression, correlated variables have to be removed; else, the model can become unstable.

Despite removing the 'amount within radius' variables, there still exists a correlation issue. Some of the 'distance to' variables, as well as the city centre distance, are correlated with each other; see the correlation plot in Figure 3. The boxes highlighted in red indicate

a correlation factor of 0.75 or higher (strong correlation). The rest of the non-significant correlations are crossed out. As such, the following variables are removed: distance to daily necessities (in favour of distance to supermarket), distances to cinema, museum, and podium (in favour of distance to nearest train station), distance to hospital and pharmacy (in favour of distance to general practitioner), distance to cafeteria (in favour of distance to cafe), and finally, as outlined in the paragraph before, distance to city centre.

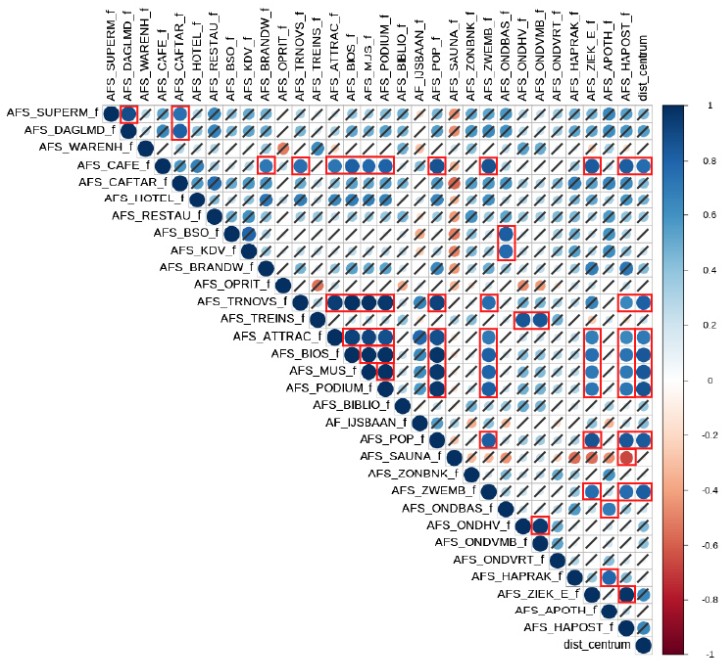

**Figure 3.** Correlation plot of 'distance to nearest…' variables of CBS (Amersfoort, 2018).

Finally, the RVO publishes a dataset containing all official energy label registrations in the Netherlands [40]. These data can be joined with the existing dataset using the ID from the BAG. This dataset also has its limitations, as not every house has an official energy label. In the past, it was not mandatory to have an energy label when selling a house. The RVO dataset only contains registrations, so not every house is present in this dataset. In addition to the energy label, the dataset also contains more detailed information on the house type and energy usage. However, due to many houses not being present in this dataset, the existing house type from Stater is used, as well as the average energy usage from CBS. In the end, the energy label is available for 70% of the houses (Table 5); for an example distribution, see Figure A2b.

The complete collection of variables is summarised in Table A4. However, there still are variables that have missing values. As has been referenced before, the number of missing values are summarised in Table 5. Here, 'Distance' refers to the distance variables of the CBS dataset. The variables not included in this overview are 100% complete. For the CBS, a large number of missing variables were resolved by also including the 500 × 500 m tiles; the number of records that uses values from the 500 × 500 m dataset is summarised in Table 6.

An additional small issue concerns the fact that not all variables are available for 2020. The most recent fully complete year is 2018. For 2020, some of the variables related to income and the 'distance to…' are not yet available. However, it is safe to assume that most of these variables have only changed a little in the last two years. As such, for 2020, we substitute the missing variables with the values from 2018.

**Table 5.** Number of missing records (% of original data), imputed using KNN (*n* = 7).

| Municipality | Build Year | Land Lot Area | Address Density | Households | Energy Usage | Distance | Energy Label |
|---|---|---|---|---|---|---|---|
| Amersfoort | 4 (0.27%) | 451 (30.19%) | 15 (1.00%) | 16 (1.07%) | 28 (1.87%) | 15 (1.00%) | 454 (30.39%) |
| Amsterdam | 116 (2.28%) | 731 (14.38%) | 0 | 71 (1.40%) | 127 (2.50%) | 0 | 1391 (27.36%) |
| Eindhoven | 16 (0.87%) | 659 (35.72%) | 0 | 93 (5.04%) | 2 (0.11%) | 2 (0.11%) | 587 (31.82%) |
| Groningen | 25 (2.16%) | 382 (32.93%) | 0 | 97 (8.36%) | 15 (1.29%) | 2 (0.17%) | 312 (26.90%) |
| Rotterdam | 1 (0.03%) | 732 (24.31%) | 0 | 39 (1.30%) | 17 (0.56%) | 0 | 948 (31.48%) |

**Table 6.** Number of observations taken from 500 × 500 m instead of 100 × 100 m.

| Municipality | WOZ-Waarde | Income |
|---|---|---|
| Amersfoort | 96 (6.43%) | 74 (4.95%) |
| Amsterdam | 398 (7.83%) | 259 (5.09%) |
| Eindhoven | 171 (9.27%) | 88 (4.77%) |
| Groningen | 138 (11.90%) | 84 (7.24%) |
| Rotterdam | 216 (7.17%) | 101 (3.35%) |

Removing all the records with missing values is not an option, as a large portion of the records have at least one or two variables missing. The result would be a dataset consisting only of a few hundred records per municipality. Instead, the unknown values are imputed from similar records. This is done using 'k-nearest neighbours imputation' with 7 neighbours. The number of neighbours is based on the fact that appraisal reports commonly use around 5 houses as reference houses. Before imputing the values, first, the variable columns are sorted from the least missing values to most missing values to guarantee that the variables with the least missing variables are imputed first.

In conclusion, four external data sources from Kadaster, CBS, and RVO are used to gather a total of 31 usable variables. The total overview of variables is presented in Table A4 in the appendix. The Kadaster mainly provides intrinsic characteristics about the house, while CBS provides the location characteristics about the neighbourhood. Additionally, RVO also provides the energy labels for a large percentage of all houses. However, not all available variables are used. Table A5 summarises the 22 variables that are not included because of high correlation with other variables or being used to derive other variables. Finally, there is the issue of missing values, as shown in Table 5. The two largest variables with missing values are the land lot areas and energy labels, which have up to 30% missing values. The missing values are imputed using 'k-nearest neighbours' with 7 neighbours to prevent throwing away the majority of records. This complete dataset is used to realise three prediction models.

### 3.4. Hyper-Parameter Optimisation Using CV

Unlike LR, GWR and XGBoost have model parameters that can be optimised. This is done using N times repeated k-folds cross-validation. In this paper, 4 folds (k = 4) are repeated 10 times (N = 10) due to the small sample size (~1k training samples) per municipality. Thus, each fold is approximately 750 samples for tuning the parameters and 250 for evaluating. Using (repeated) k-folds cross-validation reduces over-fitting and creates a better picture of the real performance. In this paper, the models are implemented using R. Specifically, using the R packages, named "lm", "GWmodel", and "xgboost", which come with built-in cross-validation methods.

For GWR, there are three parameters related to the kernel function that are fine-tuned. The kernel function itself, the kernel bandwidth, and the 'adaptive' setting. The kernel function determines the shape of the kernel. Gaussian, boxcar, and bi-square were most commonly used in the literature [26,41]. In the end, the adaptive Gaussian kernel worked best for all five municipalities. Table A3 summarises the bandwidth used by each municipality.

Finally, for XGBoost, we optimise the learning rate (eta) and the max tree depth. A higher learning rate means that the model takes larger steps towards a minimum of the loss function. The optimal learning rate lies between 0.13 and 0.17 for the five municipalities,

so they were averaged to 0.15, since the end goal is to create a single model for the entire Netherlands. This had a negligible impact on the RMSE. Similar to the tree depth, 4 out of 5 models performed best with a tree depth of 7. However, this only improved the test RMSE slightly while greatly improving the training set RMSE. As such, to prevent over-fitting, a slightly lower tree depth of 6 is chosen.

## 4. Results

This section summarises the results for the final LR, GWR, and XGBoost models that are trained. Each of the models is evaluated according to the quantitative and qualitative metrics from Section 3.1. First, the unique models for each municipality are evaluated for 2018 and 2020. Second, a single XGBoost model is evaluated that is trained on all five municipalities. Finally, a comparison is made between indexation and the five unique models, where they predict the current appraisal values of collateral's belonging to mortgages from 2000.

For the LR model, the initial model provided a poor fit mainly due to the high variance of high appraisals values. We filter out outliers above €750,000, which keeps the majority of the appraisals while making a significant improvement to the model. This is shown in the comparison of the quantile–quantile plots in Figure 4. The high appraisal values are most likely not good representatives of the total population of houses. Thus, they are excluded as they have a large influence on the prediction accuracy.

Additionally, as another alternative approach, the appraisal values were logged to model a diminishing influence of the living space. Sadly, both the log–linear model with logged appraisal values and the linear–log model with logged living spaces did not improve model accuracy. In the end, the best performing LR model is the one with the filtered out appraisal values. As summarised in Table 7, the LR model has a RMSE €85.628 and $R^2$ of 0.785, which is overall an adequate fit. Since the appraisal values vary wildly from €50,000 to €750,000, it is also worth looking at the mean absolute percentage error (MAPE) and simply the mean average error (MAE). These correspond to an average error of 9.61% and €56,219, respectively.

**Table 7.** Linear model results (Amersfoort, 2018). *: Trained using appraisals < €750,000.

| Metric | $R^2$ | RMSE | MAE | MAPE |
|---|---|---|---|---|
| LR (all appraisals) | 0.709 | €150,211 | €72,391 | 11.81% |
| LR * | 0.785 | €85,628 | €56,219 | 9.61% |
| LR-LOG * | 0.768 | €89,136 | €63,577 | 10.62% |

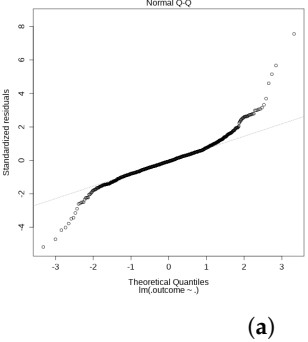

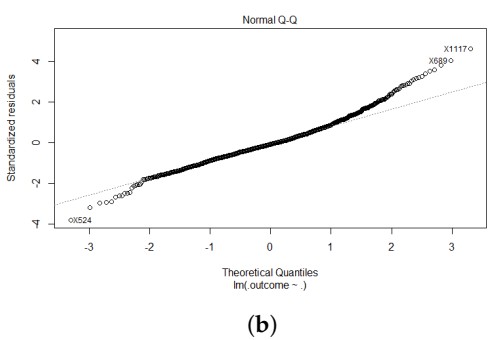

(**a**)                     (**b**)

**Figure 4.** Q–Q plot showing impact on overall fit for including all appraisals (Amersfoort, 2018). (**a**) All appraisals, poor fit. (**b**) Appraisals < €750,000, adequate fit.

The LR performance is adequate at best. Many of the CBS variables do not show a strong linear relationship with the appraisal value. Still, due to the inclusion of the living area (variable name: perceel_oppr) and WOZ-waarde, an adequate model with less than 10% deviation can still be made for Amersfoort. Figure A3 shows that these two variables

are by far the two most important factors, which is followed by the variable describing high incomes (P_HINK_HH), people aged 15–24, and build year.

The geographically weighted regression (GWR) provides a better fit than the LR model, as summarised in the GWR performance overview in Table 8. As outlined in Section 3.4, the GWR is trained using an adaptive Gaussian kernel function with varying bandwidths per municipality. For Amersfoort, the top 10 most important variables and an example of the spatial influences of the living area are plotted in Figure 5.

**Table 8.** Results of GWR models (2018).

| Municipality | $R^2$ | RMSE | MAE | MAPE |
|---|---|---|---|---|
| Amersfoort | 0.822 | €61,459 | €48,393 | 7.42% |
| Amsterdam | 0.831 | €60,213 | €53,671 | 7.31% |
| Eindhoven | 0.812 | €62,942 | €54,103 | 8.01% |
| Groningen | 0.789 | €83,233 | €55,213 | 8.61% |
| Rotterdam | 0.861 | €56,431 | €47,312 | 6.99% |

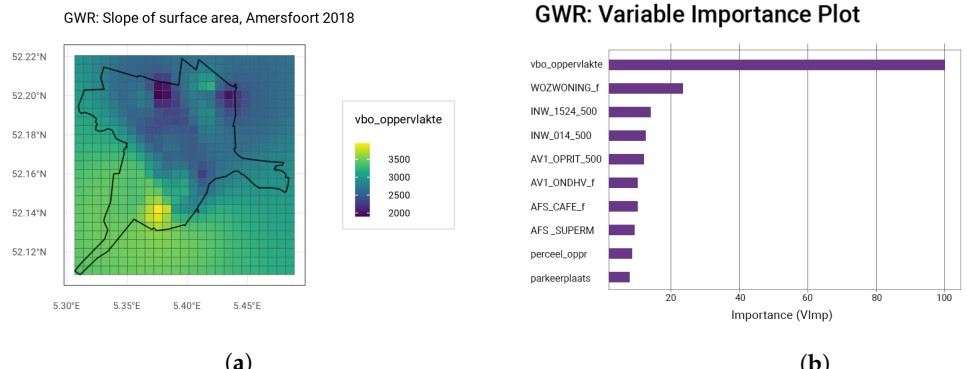

(**a**)              (**b**)

**Figure 5.** Plots describing the GWR model (Amersfoort, 2018). (**a**) The influence of living area. (**b**) Variable importance.

The most important variable is, again, the living area, which is followed by the WOZ-waarde. The variable importance plot appears to have a similar shape as the one for the linear regression (Figure A3). This time, also, some of the distance variables such as distance to nearest supermarket and cafe make an appearance. While the influence of the other variables appears to be minor, without their inclusion, the $R^2$ would be lowered by 0.09, resulting in a less good fit with an MAPE of again 10%. The final GWR manages to model the appraisal values with only 7.67% deviation on average. More important is the larger reduction of the $R^2$ and RMSE, indicating less severe outliers. The worst performing municipality is Groningen, which is likely due to it having the least samples. Rotterdam, on the other hand, performs especially well, which is perhaps due to the larger percentage of apartments in this dataset. On average, the apartments have a smaller prediction error (6.98%) than the family homes (7.41%). This can be attributed to the lower average appraisal value of apartments and lower appraisals having more reference points. The results for 2020 are summarised in Table A6 in the appendix. They show a slight decrease in predictive accuracy but not a significant one.

The final model is the XGBoost model, with parameters settings eta = 0.15, tree depth = 6, for each of the five municipalities. After 39 boosting rounds on average, no major improvements are made, and after 159 rounds, the performance starts to deteriorate slightly. The fit of the XGBoost model has the best overall fit ($R^2$ = 0.848) with the lowest RMSE scores (€58,374). A summary of the performance metrics is given in Table 9. Figure 6 shows the predicted vs. actual appraisal values for Amersfoort 2018. The other municipalities are shown in Figure A4. The living area and WOZ-waarde are again the most important variables, as seen in Figure A5. Even with the appraisals above €750,000 excluded, there is slightly more variance in the high appraisal values. Overall, the XGBoost model provides

accurate predictions with only 5% deviation on average. Table 10 summarises the average performance of each model for each of the five municipalities.

**Table 9.** Results of XGBoost models (2018).

| Municipality | $R^2$ | RMSE | MAE | MAPE |
|---|---|---|---|---|
| Amersfoort | 0.851 | €57,391 | €34,283 | 5.38% |
| Amsterdam | 0.845 | €57,964 | €35,258 | 5.50% |
| Eindhoven | 0.838 | €57,385 | €36,192 | 5.62% |
| Groningen | 0.829 | €59,832 | €38,241 | 5.88% |
| Rotterdam | 0.871 | €56,144 | €34,831 | 5.45% |

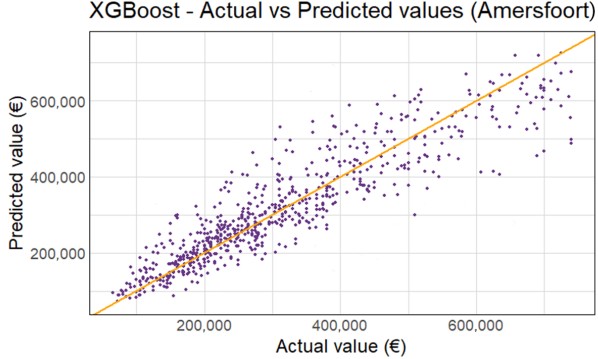

**Figure 6.** XGBoost predicted vs. actual values (Amersfoort, 2018).

**Table 10.** Averaged model performance for the five municipalities, for each model type.

| Year | 2018 | | | | 2020 | | | |
|---|---|---|---|---|---|---|---|---|
| | $R^2$ | RMSE | MAE | MAPE | $R^2$ | RMSE | MAE | MAPE |
| LR | 0.725 | €97,232 | €67,814 | 10.55% | 0.734 | €94,927 | €62,871 | 10.23% |
| GWR | 0.822 | €64,856 | €51,738 | 7.67% | 0.809 | €65,826 | €52,237 | 7.92% |
| XGBoost | 0.848 | €58,374 | €35,761 | 5.89% | 0.852 | €61,028 | €35,451 | 5.76% |

Finally, since XGBoost is the best performing model, a single XGBoost model is trained for all five municipalities using the same parameter settings (Table 11). This model includes the municipality name as an additional variable. The model's prediction error increases slightly to 6%. Furthermore, the RMSE increases substantially more than the MAE, suggesting that while the overall performance only decreased slightly, the model is worse at capturing outliers. The municipality name ends up becoming the third most important variable. While the model performance is slightly worse, it still outperforms the individually trained GWR models.

**Table 11.** Single XGBoost model trained on all five municipalities (2018).

| | $R^2$ | RMSE | MAE | MAPE |
|---|---|---|---|---|
| **XGBoost** | 0.832 | 65,312 | 43,625 | 6.35% |

All in all, when looking at the quantitative performance metrics, the XGBoost models outperform the linear regression and GWR models. The final qualitative metrics are the implementation time and model upkeep. In this research, the most effort went into gathering all the variables and preparing the dataset. As such, in practice, this is also expected to require the most maintenance. The BAG can be routinely updated using API request; however, the RVO and CBS datasets both use an extract that does not have an API endpoint. All in all, preparing the data for the model requires some handwork that cannot be easily automated.

Additionally, there is the consideration of training time. LR is simple and fast; for many millions of records, this is rarely an issue on a modern computer. On the other hand, the GWR computes regressions for a grid. In case of the municipality Amersfoort, a 100 × 100 m tile grid for Amersfoort (roughly 10 km × 10 km) equals 100 × 100 tiles = 10k tiles = 10k unique regressions that are computed. On modern hardware, this takes less than 5 min. For a national scale, the grid needs to be much larger in both dimensions; thus, the required computing power increases exponentially. Fitting the regression for the entire Netherlands likely takes a day instead of a few minutes.

Unlike GWR, XGBoost also has a GPU implementation. In this paper, the sample sizes for one year per municipality were relatively small, so even using only the CPU resulted in a good fit in less than 10 min using XGBoost. By using the GPU, XGboost is faster than the GWR model when training for the entire Netherlands. Model training time is something that does not costs much time of an employee. In the end, gathering the data and creating the dataset remains the most active time-consuming task, which takes an equal effort for all three models.

Finally, the current approach at Stater uses the Kadaster regional house price index (Table A1) to index appraisals. Both methods are compared by subtracting the indexed value from the predicted value of XGBoost, as shown in Figure 7. The two graphs are separated by the housing type, listing the predictions for all family homes and for all apartments. In both cases, XGBoost predicts higher appraisal values than the indexation method, on average €34,678 for the apartments (+17.31% higher than the index) and €28,566 (11.12%).

Two observations can be made from Figure 7. First, the XGBoost predictions for the apartments show less deviation from the index as compared to the predictions for the family homes. One explanation for this is the higher variance in the appraisal values of family homes as compared to apartments. The model is more likely to make a poor prediction for a family home than for an apartment as indicated by the larger outliers (rarely a large difference of €250k+).

Second, the difference between apartments and family homes corresponds to the other Kadaster index for housing types (Table A2). From this index, it can be seen that apartments have increased almost an additional 20% over the family homes across the entire Netherlands (2000–2020). The XGBoost model is able to account for this, whereas the regional index is not. This supports the main conclusion that the XGBoost model can be a better alternative to price indexation. An ideal index for the Kadaster would discern both region and house type. This could be a relatively simple improvement over the current method of indexation. All in all, this provides additional support to the conclusion that the model approach can be an improvement over indexation, as it is able to account for housing type.

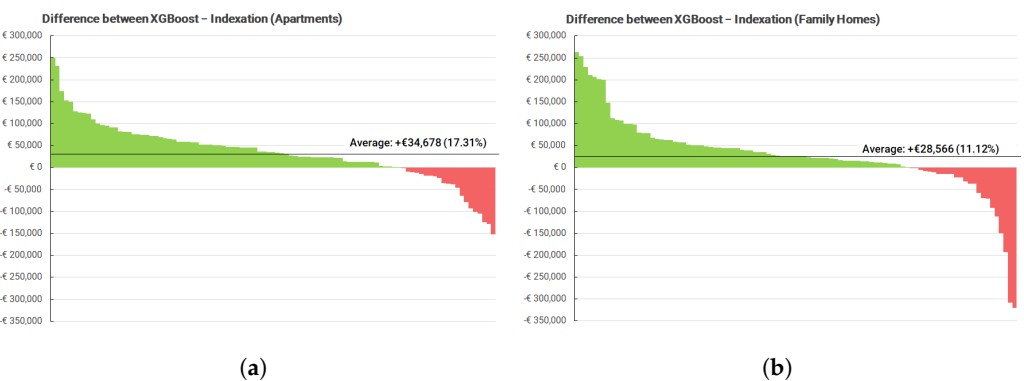

(**a**)                  (**b**)

**Figure 7.** Differences between XGBoost prediction and indexation using a regional price index (green = XGBoost predicts higher). (**a**) For apartments, XGBoost predicts 17.31% higher. (**b**) For family homes, XGBoost predicts 11.12% higher.

## 5. Discussion

In the end, the XGBoost model is able to model a large subset of the houses with a better accuracy than indexation. The model only uses the appraisal values below €750,000, since the highest appraisals (most expensive houses) caused a large increase in variance due to the stronger influence of the buyers individual preferences. This excludes only 4.24% of all appraisals. As such, the challenge to model appraisal values for the most expensive houses remains.

In the XGBoost model, the living area and taxation value (WOZ-waarde) account for 70% of the explained variance, while the other variables combined increase the explained variance by 7%. A drawback is that the WOZ-waarde is unique to the Netherlands. We argue that similar results for other countries are achievable, since the WOZ-waarde is also influenced by variables such as living area. After all, the WOZ-waarde is a rough valuation from the government. Without its inclusion, living area likely plays an even bigger role. All in all, the model has better predictive results for the Netherlands with the inclusion of the (averaged) taxation value. As shown in the comparison between indexation and XGBoost, the XGBoost is superior to indexation, as the model takes into account different types of houses (Figure 7). The remaining unexplained variance of 17% is likely due to a missing variable that explains the quality of the house. Information specific to the house from the official appraisal reports can help alleviate this variance, as they contain more information about the house itself.

In addition to XGBoost having superior accuracy compared to LR and GWR (in terms of quantitative metrics, $R^2$, RMSE and MAPE), it also performs well in terms of training time performance compared to GWR. XGBoost comes with the advantage that it can run on the GPU, whereas GWR is CPU-bound, which runs into performance problems when computing regressions for large grids of entire countries. Thus, the training time of XGBoost is not an issue when training models for all appraisal values. The largest time consumption, compared to indexation, lies in keeping the model data up-to-date, which is equally time consuming for all three models. Only the Kadaster data are easily accessible through various APIs. The CBS and RVO datasets have to be downloaded manually.

The downsides of the XGBoost model are the larger outliers compared to conservative indexation, as well as the fact that the model currently predicts for an entire year and does not account for monthly changes. This can partially be mitigated by ensuring the model gets retrained every month, replacing the appraisals of the oldest month with the new month. Finally, it takes extra effort to keep the data of the models up-to-date. However, in return for this extra effort, XGBoost can make more localised predictions for the entire Netherlands to valuate mortgage collaterals.

## 6. Conclusions

This paper investigates model-based appraisal of real estate using hedonic pricing across cities. We compare different machine learning (ML) approaches to produce hedonic pricing models, and we evaluate these on the basis of accuracy, cost, speed, and data requirements. To achieve this goal, we proposed five research questions for which we arrived at the following conclusions.

*Which ML approaches are currently used for hedonic pricing, and how do they perform?*

Four hedonic pricing models from the literature are analysed, as well as variables used in real estate value modelling. From this, we implemented three hedonic pricing models using linear regression (LR), geographically weighted regression (GWR), and extreme gradient boosting (XGBoost). They model appraisal values for five municipalities in different parts of the Netherlands: namely, Amsterdam, Amersfoort, Eindhoven, Groningen, and Rotterdam. The quantitative results for each model are presented in Table 10. The models are tested on appraisal values below €750,000, since the highest appraisals (most expensive houses) caused a large increase in variance due to the stronger influence of the buyers individual preferences.

For 2020, XGBoost best explains the variance of the appraisal values with an average $R^2$ of 0.852. This is a statistically significant improvement over GWR ($R^2 = 0.809$) and LR ($R^2 = 0.734$). For XGBoost, the mean RMSE across the five municipalities is €61,028, and the MAE is €35,451. The higher appraisal values have a larger variance than the lower appraisal values. Thus, some outliers are present in the made predictions. On average, the mean absolute percentage error (MAPE) is 5.89%. In 2020, for an average appraisal of €450,000 (in 2018), this corresponds to an error of about €27,000. Thus, XGBoost is overall a good method for modelling appraisal values.

*Which factors are significant for price differences between houses across cities? Which data are available about these factors?*

The two most important variables in all three model types are the total living area (vbo_oppervlakte, from Kadaster) and an average taxation value of all nearby houses in a $500 \times 500$ m area (WOZ-waarde, from CBS). Additionally, the other significant variables in the XGBoost model consist of the latitude of the house, percentage of incomes belonging to the 20% highest incomes in the Netherlands, electricity usage, and finally the distance to the nearest cafe. The western part of the Netherlands generally has higher appraisal values. In addition, rich people tend to live in more expensive neighbourhoods. The distance to the nearest cafe is likely related to the distance to the city centre. Other variables, such as energy labels, have little influence because they have the most missing values.

*How can we construct a method for hedonic pricing across different cities using the obtained insights? What are the results of applying this method with a realistic dataset?*

The ultimate goal is a national appraisal model for the Netherlands. The five municipalities were specifically chosen as they represent unique provinces in different parts of the Netherlands. Additionally, these municipalities have some of the largest populations. As such, we believe they provide a good mixed sample for a national model. The single XGBoost model, trained for all five municipalities, can explain 83% of the variance with an RMSE of €65,312, an MAE of €43,625, and an MAPE of 6.35% (Table 11). All in all, this XGBoost model performs only marginally worse than the five individually trained models, with only a 0.02 reduction for the $R^2$ and a 0.48% increase for the MAPE. Thus, it can be concluded that it is highly likely that XGBoost is also able to model the appraisal values for all municipalities.

Finally, a quantitative comparison between XGBoost and indexation is made by comparing the predictions of both methods for appraisal values from 2000. The predictions are discerned in two categories: apartments and family homes. In both cases, the XGBoost model makes higher predictions than the index: +17.14% for apartments and +11.12% for family homes (Figure 7). Evidently, the index is a more conservative estimate of the price increase by taking the average of many real estate prices. The predictions of the XGBoost model are also in line with the housing type index (Table A2). This index shows a larger increase of 70% in apartment prices, as compared to only 50% for family homes. This supports that the XGBoost model is able to account for differences in price development for apartments and family homes. Finally, it should be noted that the XGBoost model also has a few outliers in its predictions for the family homes. However, based on the training results for 2018, it can be concluded that the XGBoost model can be more reliable than indexation for the majority of appraisals, excluding the most expensive appraisals.

Based on the previous conclusions, we come to the following recommendations for future research centred around modelling real estate values using open data and XGBoost:

- The lack of a feature to model house quality. The remaining unexplained variance of 17% is likely due to a missing variable that explains the quality of the house itself or other location characteristics. An official appraisal report contains more detailed information about the state of a house. This can help paint a better picture of the house itself.
- For example, the ground sinkage map from TU Delft provides an interesting use case for looking at real estate portfolio risk factors. Ground sinkage is a real problem

in the Netherlands, especially in Groningen. As a result of the gas exploitation, the property values are reduced drastically in the region. This poses a clear risk to the mortgage owner and the money lender. Another problem for many houses is foundation rot; perhaps risk areas can be identified by combining sinkage data with ground compositions.

**Author Contributions:** Conceptualisation, Evert Guliker and Erwin Folmer; methodology, Evert Guliker; software, Evert Guliker; validation, Evert Guliker, Erwin Folmer and Marten van Sinderen; formal analysis, Evert Guliker; investigation, Evert Guliker; resources, Evert Guliker; data curation, Evert Guliker; writing—original draft preparation, Evert Guliker; writing—review and editing, Evert Guliker, Erwin Folmer and Marten van Sinderen; visualisation, Evert Guliker; supervision, Erwin Folmer and Marten van Sinderen; project administration, Erwin Folmer and Marten van Sinderen. All authors have read and agreed to the published version of the manuscript.

**Funding:** This research received no external funding.

**Institutional Review Board Statement:** Not applicable.

**Informed Consent Statement:** Not applicable.

**Data Availability Statement:** Publicly available datasets were used in this study. Sources: CBS, for the socio-demographic variables [39]; Kadaster, for the build year, living area (BAG: [9]), and land lot area (DKK [38]); RVO, for the energy labels [40]. The appraisal data were provided by Stater N.V. and are not publicly available due to the privacy concerns.

**Conflicts of Interest:** The authors declare no conflict of interest.

## Abbreviations

The following abbreviations are used in this manuscript:

| | |
|---|---|
| LR | Linear regression |
| (M)GWR | (Multi-scale) Geographically weighted regression |
| XGBoost | Extreme gradient boosting |
| CBS | 'Centraal Bureau voor de Statistiek' (ENG: Central Agency for Statistics) |
| BAG | 'Basisregistratie adressen & gebouwen' (ENG: Base registry addresses & buildings) |
| DKK | 'Digitale kadastrale kaart' (ENG: Digital cadastral map) |

## Appendix A. Figures Related to Dutch Housing Market

**Table A1.** % change in house prices (January 2000–January 2020), per provinces of the Netherlands [42].

| Province | % Increase over 2000–2020 | Province | % Increase over 2000–2020 |
|---|---|---|---|
| Drenthe | 56.35% | Noord-Brabant | 42.90% |
| Flevoland | 44.02% | Noord-Holland | 76.70% |
| Friesland | 55.34% | Overijssel | 49.73% |
| Gelderland | 45.05% | Utrecht | 70.87% |
| Groningen | 67.48% | Zeeland | 74.83% |
| Limburg | 38.16% | Zuid-Holland | 52.36% |

**Table A2.** % change in house prices (January 2000–January 2020), per housing type. Source: Kadaster [42].

| Housing Type | % Increase over 2000–2020 |
|---|---|
| Detached | 54.4% |
| Semi-detached | 51.2% |
| Terraced House | 64.0% |
| Corner House | 61.5% |
| Apartment | 75.3% |

## Appendix B. Figures Related to Models

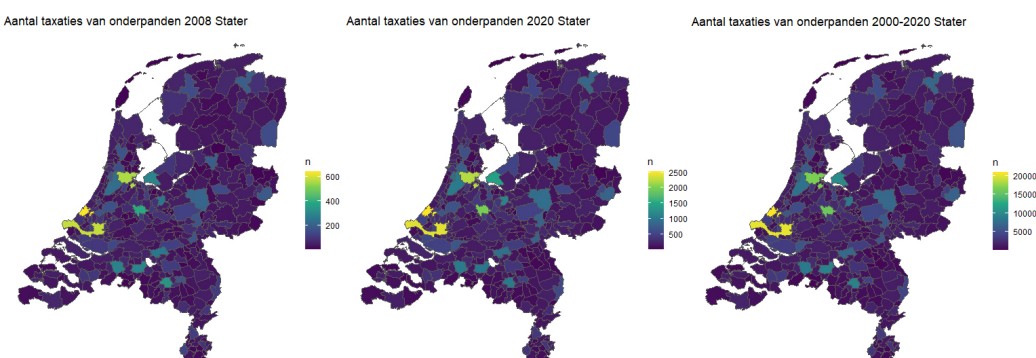

**Figure A1.** Number of real estate appraisals of Stater, (**left**) 2008, (**middle**) 2020, (**right**) January 2000–January 2021.

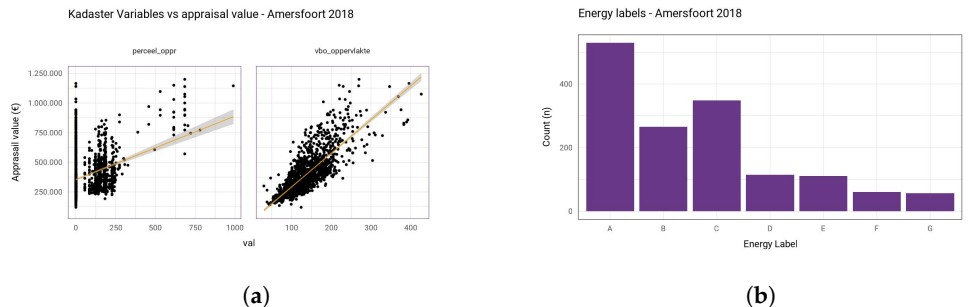

(**a**)    (**b**)

**Figure A2.** Exploration of external variables from Kadaster & CBS (Amersfoort, 2018). (**a**) Kadaster—Land lot size ($m^2$) & total floor area ($m^2$). (**b**) RVO—Energy Labels.

**Table A3.** Best kernel settings for GWR model (2018).

| Municipality | Kernel (Bandwidth) | Adaptive/Fixed |
|---|---|---|
| Amersfoort | Gaussian (0.28) | Adaptive |
| Amsterdam | Gaussian (0.19) | Adaptive |
| Eindhoven | Gaussian (0.27) | Adaptive |
| Groningen | Gaussian (0.43) | Adaptive |
| Rotterdam | Gaussian (0.25) | Adaptive |

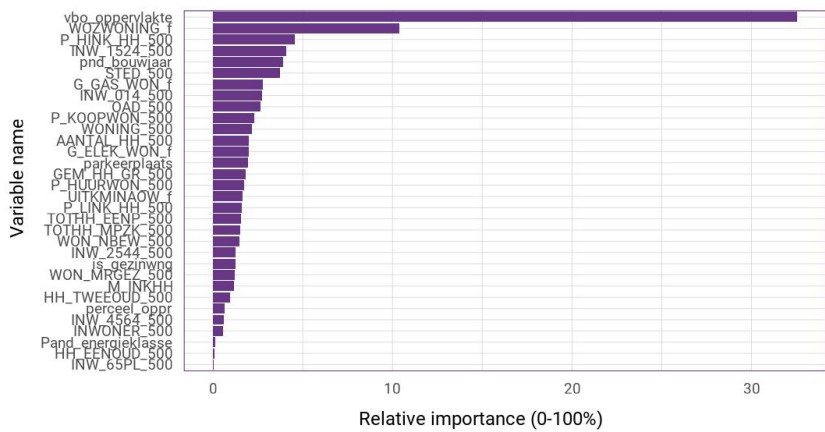

**Figure A3.** Variable importance for the LR model of Amersfoort (2018). All 5 municipalities have similar results.

**Table A4.** Variables of final models and their variable inflation factors (VIF) (Amersfoort, 2018).

| Abbreviation | Description | VIF | Source |
|---|---|---|---|
| is_gezinwng | Apartment (0) or Family home (1) | 1.91 | Stater |
| garage | Presence of garage (yes/no = 1/0) | 1.82 | - |
| parkeerplaats | % of people aged 0–14 (500 m tiles) | 1.43 | - |
| vbo_oppervlakte | The total floor area of a house (m$^2$) | 1.82 | Kadaster |
| pnd_bouwjaar | Build year | 2.31 | - |
| perceel_oppr | The total floor area of a house (m$^2$) | 1.80 | - |
| Pand_energieklasse | Energy label / class (factor) | 2.69 | RVO |
| INW_014 | % of people aged 0–14 (500 m tiles) | **15.05** | CBS |
| INW_1524 | % of people aged 15–24 (500 m tiles) | 1.53 | - |
| INW_2544 | % of people aged 25–44 (500 m tiles) | **11.47** | - |
| INW_4464 | % of people aged 45–64 (500 m tiles) | **7.67** | - |
| INW_65PL | % of people aged 65+ (500 m tiles) | **10.91** | - |
| TOTHH_EENP | % of single person house holds | 4.12 | - |
| TOTHH_MPZK | % of households > 1 and no children | 4.78 | - |
| HH_EENOUD | % of one parent households with children | 4.68 | - |
| WON_MRGEZ | % of family homes | 4.4 | - |
| WON_NBEW | % non-inhabited homes | 1.90 | - |
| OAD | Address density (address/km$^2$) | 3.87 | - |
| STED_500 | Urbanisation (factor) | **6.12** | - |
| P_KOOPWON | % owner-occupied home | 3.43 | - |
| WOZWONING | Average WOZ-Waarde (×1000€) | 3.15 | - |
| M_INKHH | Median income group (factor) | 4.12 | - |
| G_ELEK_WON | Average Electricity Usage (kwH) | 2.10 | - |
| P_LINK_HH | % of households belonging to bottom 40% of national income | **13.12** | - |
| P_HINK_HH | % of households belonging to top 20% of national income | **14.45** | - |
| AFS_SUPERM | Distance to nearest supermarket (km) | 3.22 | - |
| AFS_OPRIT | Distance to nearest provincial road or highway (km) | 2.48 | - |
| AFS_CAFE | Distance to nearest cafe (km) | 2.21 | - |
| AFS_BIBLIO | Distance to nearest library (km) | 2.27 | - |
| AFS_ONDVRT | Distance to nearest secondary education (km) | 1.77 | - |
| AFS_APOTH | Distance to nearest pharmacy (km) | 2.08 | - |

**Table A5.** Variables excluded due to high correlation with other variables.

| Abbreviation | Description | Source |
|---|---|---|
| dist_centre | Distance to city center (km) | Self-computed |
| UITKMINAOW | Income from state pension (AOW) | CBS |
| INWONER | Inhabitants at start of year | - |
| AANTAL_HH | Number of households. | - |
| HH_TWEEOUD | % of two parent households with children | - |
| P_NW_MIG_A | Percentage of inhabitants (non-western) | - |
| P_HUURWON | Percentage of rented homes | - |
| G_GA_WON | Average Gas Usage (m$^3$) | - |
| AV1/5/10/20 vars. | Variables describing 'Amount of X within radius 1/5/10/20 km' (hospitals, stores, schools etc.) | - |
| Other AFS vars. | Distance variables to other amenities (Swimming pool, attraction parks, restaurants, hotels, hospital and others.) | - |
| Pand_gebouwtype | Home type | RVO |
| Pand_subtype | Home subtype | - |

**Table A6.** Results for GWR models (2020).

| Municipality | $R^2$ | RMSE | MAE | MAPE |
|---|---|---|---|---|
| Amersfoort | 0.810 | €61,928 | €50,177 | 7.51% |
| Amsterdam | 0.822 | €62,596 | €52,183 | 7.40% |
| Eindhoven | 0.815 | €62,942 | €54,631 | 7.98% |
| Groningen | 0.821 | €79,192 | €54,131 | 8.29% |
| Rotterdam | 0.837 | €58,561 | €49,287 | 7.25% |

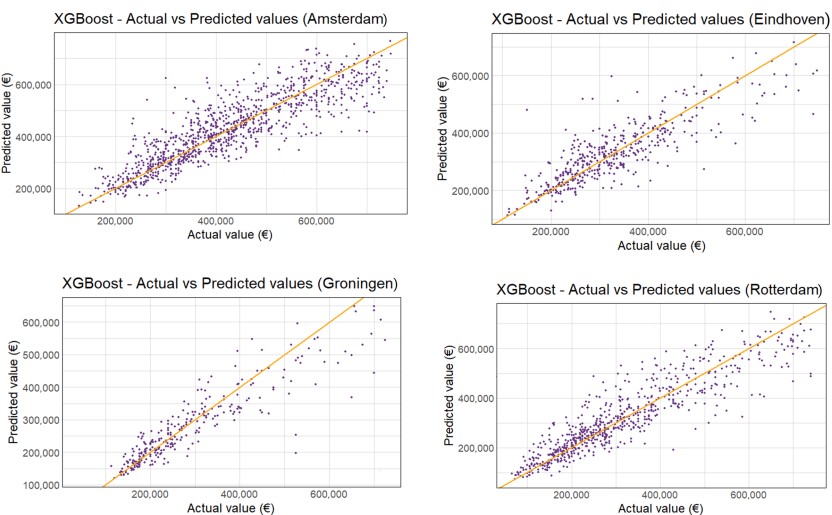

**Figure A4.** Model fit of XGBoost models for Amsterdam, Eindhoven, Rotterdam, Groningen (2018), (orange line is y = x).

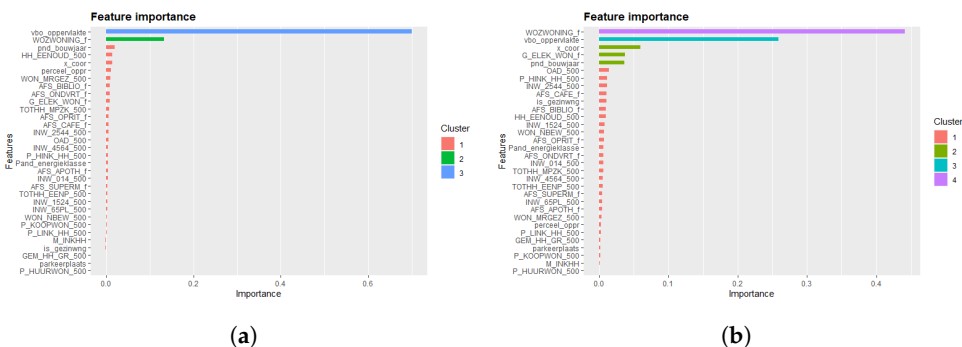

(**a**)             (**b**)

**Figure A5.** XGBoost Variable Importance of Amersfoort & Amsterdam (2018). (**a**) Amersfoort. (**b**) Amsterdam.

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
