# Peer review of "Spatial Determinants of Real Estate Appraisals in The Netherlands: A Machine Learning Approach"

_ijgi, doi:10.3390/ijgi11020125_

Round 1
Reviewer 1 Report
The goal of this paper is to investigate model-based appraisal of real estate using hedonic pricing across cities and publicly available data. Authors wanted to compare different machine learning (ML) approaches to produce hedonic pricing models, and evaluate these on basis of accuracy, cost, speed and data requirements.
The article is well structured, well written and clearly presents its methods and results. English language and style minor spell check required.
Author Response
Please see the attachement (reviewer 1)

Reviewer 2 Report
Some comments for authors follow.
1) In the Abstract, the abbreviation XGBoost is introduced without the corresponding meaning. Moreover, references to Tables and Figures are not useful in the Abstract: please, remove them.
2) Line 70: change "an traditional" to "a traditional"
3) Lines 84-85: It is not clear what the authors want to say in these two sentences.
4) Change everywhere in the manuscript "in literature" to "in the literature"
5) Line 137: change "They maintains" to "It maintains"
6) In several points of the manuscript there are references to Tables and Figures put in the Appendix, but without specified that those Tables/Figures are in the Appendix. Evidently, this causes confusion to the readers.
7) Lines 169-171: the sentence "The most....negative value." is not clear
8) Lines 255-256: adjust the sentence "The consequence....given variables."
9) Figure 1-c) of page 10: correct the label of x-axis
10) Line 460: I think the reference must be to Table 5 and not Figure 5.
11) Lines 499, 521, 522, 528, 580: there are references to Tables/Figures/section without any specification if the number refers to a table, a figure or a section.
12) Line 529: change "all variable are" to "all variables are"
13) Line 581: change "Table 10 summarise" to "Table 10 summarizes"
14) Line 639: there isn't any Figure 11 in the manuscript.
15) Line 647: change "XGBoost models outperforms" to "XGBoost models outperform"
16) Line 705: delete "for," at the beginning of the line
Author Response
Please see the attachement (reviewer 2)

Reviewer 3 Report
Review on the paper entitled “Spatial Determinants of Real Estate Appraisals in the Netherlands: a Machine Learning Approach”.
The main goal of the paper is to investigate model-based appraisal of real estate using hedonic pricing across cities and publicly available data. It is a well-organized and high quality paper with strong scientific content. I have only some minor remarks that the Authors should address.
Lines 79-80, page 2: The Authors mention the name of Calcasa but the readers are not provided with information what Calcasa is. At least the industrial profile of the company (real estate company, software company, etc.) should be clarified.
It is not entirely clear what the difference between “strongly” and “highly positive” is (see the terms in Table 2).
Line 522, page 12: I think something is missing from this sentence: “The complete collection of variables is summarised in 3.”
Lines 712-720, page 19: The Authors list “the most important variables” of the three model types, and the top-5 most important ones of the XGBoost model. I couldn’t find a paragraph, or a Table/Figure in the previous chapters summarizing those variables especially in the case of the XGBoost model. There are many variables listed in Table 3 in the Appendix, but I am not sure whether the Authors wanted to refer to those variables on page 19. In addition, the Authors should clarify what “x-coordinate (horizontal position)” means.
Author Response
Please see the attachment (reviewer 3)

This manuscript is a resubmission of an earlier submission. The following is a list of the peer review reports and author responses from that submission.